# Expression of NK Cell Receptor Ligands on Leukemic Cells Is Associated with the Outcome of Childhood Acute Leukemia

**DOI:** 10.3390/cancers13102294

**Published:** 2021-05-11

**Authors:** María Victoria Martínez-Sánchez, José Luis Fuster, José Antonio Campillo, Ana María Galera, Mar Bermúdez-Cortés, María Esther Llinares, Eduardo Ramos-Elbal, Juan Francisco Pascual-Gázquez, Ana María Fita, Helios Martínez-Banaclocha, José Antonio Galián, Lourdes Gimeno, Manuel Muro, Alfredo Minguela

**Affiliations:** 1Immunology Service, Clinic University Hospital Virgen de la Arrixaca and Biomedical Research Institute of Murcia (IMIB), 30120 Murcia, Spain; mariav.martinez3@carm.es (M.V.M.-S.); josea.campillo@carm.es (J.A.C.); helios.martinez2@carm.es (H.M.-B.); josea.galian3@carm.es (J.A.G.); lgarias@um.es (L.G.); manuel.muro@carm.es (M.M.); 2Pediatric Oncohematology Department, Clinic University Hospital Virgen de la Arrixaca and Biomedical Research Institute of Murcia (IMIB), 30120 Murcia, Spain; josel.fuster@carm.es (J.L.F.); anam.galera@carm.es (A.M.G.); mariam.bermudez2@carm.es (M.B.-C.); mariae.llinares@carm.es (M.E.L.); eduardo.ramos2@carm.es (E.R.-E.); juanf.pascual.sspa@juntadeandalucia.es (J.F.P.-G.); anam.fita.sspa@juntadeandalucia.es (A.M.F.); 3Human Anatomy Department, University of Murcia (UM), 30100 Murcia, Spain

**Keywords:** NK cells, KIR, HLA, childhood acute leukemia

## Abstract

**Simple Summary:**

Natural killer cells (NK cells) of the innate immune system are suspected of playing an important role in eliminating residual leukemia cells during maintenance chemotherapy given to children with acute lymphoblastic leukemia for about two years. This study analyzes the expression of ligands for the receptors that regulate the function of NK cells on leukemic cells of more than one hundred children with acute lymphoid and myeloid leukemia. Our results show that the loss of expression of some molecules involved in the activation of NK cells is associated with poorer survival. In addition, a genetic combination of molecules that interact to regulate NK cell function seems to be associated with a higher relapse rate during/after chemotherapy and shorter patient survival. Children who carry this genetic combination are refractory to current chemotherapy treatments, and stem cell transplantation does not seem to contribute to their cure either, and therefore, they should be considered as candidates for alternative biological therapies that might offer better results.

**Abstract:**

Acute leukemia is the most common malignancy in children. Most patients are cured, but refractory/relapsed AML and ALL are the first cause of death from malignancy in children. Maintenance chemotherapy in ALL has improved survival by inducing leukemic cell apoptosis, but immune surveillance effectors such as NK cells might also contribute. The outcome of B-ALL (*n* = 70), T-ALL (*n* = 16), and AML (*n* = 16) pediatric patients was evaluated according to leukemic cell expression of ligands for activating and inhibiting receptors that regulate NK cell functioning. Increased expression of ULBP-1, a ligand for NKG2D, but not that of CD112 or CD155, ligands for DNAM-1, was associated with poorer 5-year event-free survival (5y-EFS, 77.6% vs. 94.9%, *p* < 0.03). Reduced expression of HLA-C on leukemic cells in patients with the KIR2DL1/HLA-C*04 interaction was associated with a higher rate of relapse (17.6% vs. 4.4%, *p* = 0.035) and lower 5y-EFS (70.6% vs. 92.6%, *p* < 0.002). KIR2DL1/HLA-C*04 interaction was an independent predictive factor of events (HR = 4.795, *p* < 0.005) or death (HR = 6.731, *p* < 0.005) and might provide additional information to the current risk stratification. Children who carry the KIR2DL1/HLA-C*04 interaction were refractory to current chemotherapy treatments, including allogeneic stem cell transplantation; therefore, they should be considered as candidates for alternative biological therapies that might offer better results.

## 1. Introduction

Acute leukemia is the most common malignancy in children and is characterized by the accumulation of leukocyte progenitors in the bone marrow (BM) [1]. In the last two decades, great progress has been made in the biological understanding of the disease, thus promoting molecular diagnosis, patient risk stratification, therapy personalization, early evaluation of treatment response, and substantial improvement in disease management and clinical outcome. Nonetheless, current treatment results of childhood acute myeloblastic leukemia (AML) are considered sub-optimal and new drugs are being evaluated [1]. In contrast, most children with acute lymphoblastic leukemia (ALL) are cured with current multi-agent chemotherapy regimens, including in most patients a maintenance phase of less intensive chemotherapy for a prolonged period of time, which is critical for ensuring the cure of the disease [2]. Despite these encouraging advances, the outcome of patients with refractory/relapsed AML and ALL is dismal, and these types of leukemia still are the most common cause of death from malignancy in children [3]. The ALL therapeutic landscape has changed dramatically with the incorporation of novel immunotherapies, such as the bispecific T-cell engager (BiTE) that links CD19^+^ B-cells with CD3^+^ T cells, the calicheamicin-conjugated anti-CD22 monoclonal antibody, and multiple CD19-targeted chimeric antigen receptor (CAR) T cells [4,5,6,7]. Natural killer (NK) cell therapies are also offering new opportunities to patients with refractory AML and ALL, either through hematopoietic stem cell transplantation (HSCT) with donors selected to maximize NK-cell alloreactivity [8,9] or through ex vivo activation of patients’ own NK cells or allogeneic NK cells [10,11]. To improve the efficacy of these treatments [12], advances in the knowledge of the interactions that govern NK cell function are needed, particularly those interactions promoted by molecules expressed on leukemic cells that could define the type and potency of NK cell antitumor response.

Antileukemic NK cell response is regulated by the integration of signals from activating FcγRIII, NKG2D, DNAM-1, NKp30, NKp44, or NKp46 receptors [13] and inhibitory NKG2A or killer-cell immunoglobulin-like receptors (KIR) when interacting with their specific ligands on the transformed cells. The balance between positive and negative signals allows NK cells to differentiate healthy tissues from tissues expressing damage/danger ligands for activating receptors or from tissues with loss of ligands for inhibitory receptors such as human leukocyte antigens class I (HLA-I), also known as missing-self [14,15]. The loss of HLA-I expression to escape the cytolytic activity of T lymphocytes is a frequent event in cancer [16], and it is in these circumstances that the antitumor function of NK cells becomes particularly relevant [17]. However, NK cells are also able to sense alterations in the peptidome presented by specific HLA allotypes that may occur during viral infections or malignant transformations [18,19]. For this reason, the patient’s genetic profile, particularly the profile of the highly polymorphic KIR/HLA-I interactions, can condition the immune response, and thus, the appearance and progression of malignant diseases.

To better understand NK cell functioning, we must take into account that during their development, the recognition of their own HLA-I by NKG2A and inhibitory KIR (iKIR) receptors promotes their maturation and functional competence, educating them in a process called “licensing”. This increases the expression of the DNAM-1 activating receptor and the stability of NK/tumor cell synapse [20]. On the contrary, the lack of self-recognition through inhibitory receptors leads to incomplete maturation and hyporesponsive NK cells [21,22,23]. Inhibitory licensing interactions include KIR2DL1/C2-epitope (HLA-C allotypes with Lys80 in alpha-1 helix), KIR2DL2-3/C1-epitope (HLA-C allotypes with Asn80 and the rare allotypes HLA-B*46:01 and HLA-B*73:01) [18,24], KIR3DL1/Bw4-epitope (HLA-A and HLA-B allotypes with Bw4 epitope, either with Thr80 -80T- or Ile80 -80I-) [25], KIR3DL2/HLA-A*03 and A*11 allotypes [26], and NKG2A/HLA-E [27]. KIR2DL2-3 can also interact with C2-epitope, although with lower affinity [18].

Interactions for activating KIRs (aKIR) have been reported for KIR2DS1/HLA-C2 [28,29], KIR2DS2/C1 [30], KIR2DS2/HLA-A*11 [31], KIR2DS2/non-HLA ligands expressed in cancer [32], KIR2DS4/HLA-A*11, C*01:02, C*02:02, C*04:01, C*05:01, C*14:02, and C*16:01 [33]. KIR3DS1 ligands have remained indefinite until recently, and they are represented by the open conformation of the non-classical HLA-F molecule and by the HLA-B*51 (an HLA-B Bw4 80I allotype) [18,34,35]. KIR2DS5*006 (present in Africans) recognizes several HLA-C2 molecules [36]. Unlike iKIRs, which educate and enhance the effector activity of NK cells, signaling through aKIRs induces hyporesponsiveness [28,29]. In fact, aKIR/HLA interactions seem to have detrimental effects on immunosurveillance against hematopoietic malignancies, with patients showing increased susceptibility to cancer and/or worse clinical outcomes [37,38,39]. However, these results are contradictory with others that report survival benefits in patients with AML who receive donor grafts with KIR B-haplotypes, rich in aKIRs [40].

Although precise mechanisms have not yet been elucidated, education via aKIRs shares features with the hyporesponsiveness induced by chronic stimulation through the activating receptor NKG2D when interacting with its ligands, the major histocompatibility complex class I chain-related proteins A and B (MICA/B), and the UL16-binding protein (ULBP)-1. Sustained interaction of NKG2D receptor with its ligands expressed on tumor cells has been reported to decrease the association of NKG2D with DAP-10 and KARAP/DAP-12 adaptor proteins by uncoupling NKG2D receptor signaling from intracellular calcium mobilization and decreasing NK cell-mediated cytolysis [41]. Nonetheless, efficient tumor cell killing requires additional activating signaling, such as that of DNAM-1 (CD226) interacting Nectin-2 (CD112) and poliovirus receptor (PVR or CD155) ligands. Although the clinical implications of these interactions have been poorly studied, alterations in DNAM-1 expression lead to higher relapse rates in AML [42] and higher susceptibility to solid and hematologic cancers [43,44,45,46].

NK cells exhibit a multitude of receptors capable of transmitting activating and inhibitory signals when interacting with their specific ligands [47], and therefore, a holistic approach is necessary for a better understanding of their antitumor response. In this study, however, we analyzed the expression of ligands for NK cell receptors that, according to our own experience, play a key role in NK cell education and immune surveillance of different types of cancer [48,49,50,51]. For that reason, the expression of HLA-I and HLA-C, main ligands of iKIRs and aKIRs, and that of CD112/CD155 and ULBP-1/MICA/B, ligands of CD226 and NKG2D, respectively, were prospectively analyzed on B-ALL, T-ALL, and AML pediatric leukemic cells to ascertain their role in the outcome of these patients.

## 2. Results

### 2.1. Characteristics and Outcome of Patients According to the Type of Acute Leukemia

Table 1 summarizes the biological and clinical characteristics as well as treatment and outcome of patients according to the type of acute leukemia. Mean follow-up was 40.0 ± 31.3 months. Treatment-related mortality and cumulative incidence of relapse were 8.6% and 2.8% for B-ALL, 0.0% and 12.5% for T-ALL, and 6.25% and 31.2% for AML. Five-year event-free survival (5y-EFS) and overall survival (5y-OS) rates were 88.6% and 90.0% for B-ALL, 87.5% and 100% for T-ALL, and 68.8% and 68.8% for AML, respectively. According to the risk status at enrollment, treatment-related mortality and cumulative incidence of relapse were 7.7% and 0.0% for standard-risk, 2.5% and 2.5% for intermediate-risk, and 11.8% and 20.6% for high-risk patients; 5y-EFS and 5y-OS were 92.3% and 92.3% for standard-risk, 95.0% and 97.5% for intermediate-risk, 67.6% and 73.5% for high-risk patients (Figure 1A). Detailed results for each type of leukemia according to patient risk can be seen in Figure 1B.

### 2.2. Patient Outcome According to the Expression of Ligands for NK Cell Receptors on Leukemic Blast Cells

First, we evaluated the expression of ligands for activating NK cell receptors measured as mean fluorescence intensity (MFI) on normal and leukemic blast cells (Figure 2A). Compared with their normal residual counterparts (normal lymphocytes for ALL and normal granulocytes for AML), leukemic blast cells showed similar levels of MICA/B expression in B-ALL (123.0 ± 13.1 vs. 135.9 ± 11.1 MFI, n.s.), and T-ALL (119.5 ± 34.0 vs. 172.6 ± 37.6 MFI, n.s.), but lower in AML (394.1 ± 49.3 vs. 316.9 ± 120.1 MFI, *p* = 0.013); higher levels of ULBP-1 in B-ALL (73.8 ± 10.0 vs. 263.1 ± 42.5 MFI, *p* < 0.001), T-ALL (126.7 ± 38.5 vs. 251.5 ± 46.8 MFI, n.s.), and AML (636.2 ± 119.6 vs. 719.2 ± 300.0 MFI, *p* = 0.046); higher levels of CD112 in B-ALL (340.8 ± 48.8 vs. 1868 ± 140.6 MFI, *p* < 0.001), T-ALL (350.4 ± 84.4 vs. 914.3 ± 245.0 MFI, n.s.), and AML (13,154 ± 345 vs. 3422 ± 476.5 MFI, *p* < 0.001); and higher levels of CD155 in B-ALL (99.1 ± 6.0 vs. 417.8 ± 128.0 MFI, *p* < 0.001), and T-ALL (170.1 ± 44.2 vs. 415.0 ± 80.3 MFI, *p* = 0.026), but similar in AML (1029 ± 137.2 vs. 946.2 ± 114.5 MFI, n.s.).

When comparing the expression of ligands for inhibitory NK cell receptors with that of their normal counterpart cells (Figure 2B), leukemic blast cells showed higher expression of total HLA-I in B-ALL (38,277 ± 3708 vs. 45,314 ± 4975 MFI, n.s.) and AML (37,187 ± 9152 vs. 51,888 ± 9623 MFI, n.s.) but lower in T-ALL (37,122 ± 7468 vs. 17,559 ± 4387 MFI, *p* = 0.046); and lower expression of HLA-C in B-ALL (14,320 ± 1489 vs. 10,474 ± 1212 MFI, *p* = 0.028) and T-ALL (12,412 ± 2509 vs. 6861 ± 1832 MFI, *p* = 0.019) but similar levels in AML (7828 ± 1453 vs. 8877 ± 1523 MFI, n.s.). However, when studying patients individually, we detected lower expression (loss of HLA) on leukemic blasts compared to their normal counterpart cells for HLA-I in 42.4%, 93.3%, and 25.0% (*p* = 0.0002) and HLA-C in 75.8%, 86.7%, and 62.5% (*p* = 0.292) of B-ALL, T-ALL, and AML patients, respectively (Figure 3A).

Next, we explored the impact of NK cell receptor ligand expression on outcome of acute leukemia patients. Reduced 5y-EFS rate was observed for patients showing loss of HLA-I (78.3% vs. 90.2%, *p* = 0.115) and HLA-C (80.8% vs. 95.8%, *p* = 0.091) on leukemic blast cells compared to patients with maintained or increased expression of these ligands, although these results were not statistically significant (Figure 3B). Regarding ligands for activating NK cell receptors, reduced 5y-EFS rate was observed for patients showing high vs. low expression (cutoff = 140.1) of ULBP-1 (77.6% vs. 94.9%, *p* = 0.03). However, there were no significant differences in the 5y-EFS rate of patients showing high or low expression of MICA/B (cutoff = 108.5; 79.7% vs. 92.1%, *p* = 0.126), CD112 (cutoff = 1640; 84.5% vs. 84.6%, *p* = 0.903), or CD155 (cutoff = 326.1; 83.6% vs. 86.1%, *p* = 0.652) (Figure 3C).

### 2.3. Role of KIR/HLA-Ligand Interactions in the Outcome of Acute Leukemia

To investigate the role of NK cell receptor interactions with their ligands expressed on leukemic blast cells, we first explored the role of HLA-I allotypes in the susceptibility to childhood acute leukemia by analyzing their frequency in control and patient groups (Appendix A). Only HLA-B*35 was significantly more frequent in childhood acute leukemia than in controls (34.0% vs. 12.0%, *p* = 0.0005, Pc = 0.009). Since HLA-B*35 is not a Bw4 allele and its interaction with any NK cell receptor has not been reported, we will evaluate its potential role in patient outcome in the following sections of this manuscript. HLA-C*04 also showed higher frequency in patients than in controls (33.3% vs. 20.5%, *p* = 0.037, Pc = 0.185) but its statistical significance was lost after correction.

Next, we evaluated the frequency of inhibitory and activating KIR receptors, HLA-I ligands, and specific KIR/HLA-I ligand interactions in controls and in total patients, as well as in B-ALL, T-ALL, and AML patients independently (Table 2). No differences in the frequency of KIR genes, HLA-I ligands (HLA-C1, -C2, -A*03/A*11, -Bw4, Bw4-80T, Bw4-80I, or KIR2S4-ligands) or KIR/HLA-I ligand interactions were found between controls and patients with any type of acute leukemia, except for a higher frequency of KIR2DS4-ligands in total patients than in controls (71.0% vs. 53.0%, *p* = 0.009, Pc = 0.045).

Finally, we evaluated the impact of KIR/HLA-I ligand interactions in the outcome of acute leukemia patients. Specifically, only the presence of the KIR2DL1/C2-ligand interaction was associated with lower 5y-EFS rates (80.0% vs. 96.5%, *p* = 0.032) in total patients (Figure 4A left, B), as well as in B-ALL (84.8% vs. 95.8%, n.s.), T-ALL (80.0% vs. 100%, n.s.) and AML (64.3% vs. 100%, n.s.) (Figure 4B). To clarify the role of this interaction, we subsequently analyzed the interaction of KIR2DL1 with each of the HLA-C allotypes included in the C2-epitope group (HLA-C*02, *04, *05, *06, *15, *17, *18) (Figure 4A right). It was noteworthy that specifically KIR2DL1/HLA-C*04 interaction was associated with the most reduced 5y-EFS rate (70.6% vs. 92.6%, *p* < 0.002) and 5y-OS rate (73.5% vs. 95.6%, *p* = 0.001) in all patients (Figure 4C). Furthermore, KIR2DL1/HLA-C*04 interaction was associated with a higher rate of relapse (17.6% vs. 4.4%, *p* = 0.035) compared to patients without this interaction. Patients with the KIR2DL1/HLA-C*04 interaction showed the most reduced 5y-EFS rates also in B-ALL (77.3% vs. 93.8%, *p* = 0.036), T-ALL (75.0% vs. 91.7%, n.s.) and AML (50.0% vs. 87.5%, *p* = 0.05) patients. Apparently, the KIR2DL1/HLA-C*05 interaction was associated with a better 5y-EFS rate (100% vs. 82.8%), neutralizing the negative impact of KIR2DL1/HLA-C*04 interaction. Thus, only in the absence of HLA-C*05, KIR2DL1/HLA-C*04 interaction showed the most reduced 5y-EFS rate (60.0%, *p* = 0.0001) compared to that of patients with the KIR2DL1/HLA-C*04 interaction in the presence of HLA-C*05 (100%) or to those without the KIR2DL1/HLA-C*04 interaction whether (96.0%) or not (90.7%) patients were HLA-C*05 (Figure 4D).

In Section 2.3 of this manuscript, we described that HLA-B*35 was significantly associated with acute leukemia. However, as shown in Appendix A, it was mainly due to its association with HLA-C*04, which may also result in HLA-B*35 patients having a poorer outcome due to the KIR2DL1/HLA-C*04 interaction. Furthermore, HLA-B*35 did not add any independent prognostic value to the EFS of patients (HR = 1.148, *p* = 0.837) when analyzed together with risk status at enrollment (HR = 2.751, *p* = 0.016) and KIR2DL1/HLA-C*04 interaction (HR = 4.239, *p* = 0.039, Harrell C-statistic = 0.804 ± 0.052).

Trying to understand why KIR2DL1/HLA-C*04 interaction could be associated with a much lower EFS and OS, the expression of total HLA-I and HLA-C was analyzed on leukemic blast cells of patients carrying the 6 different allotypes with the C2-epitope. Remarkably, only patients with the HLA-C*04 allotype, compared to patients with other C2-epitope allotypes, showed a significant reduction in HLA-C expression on leukemic cells (7391 ± 1046 vs. 10,765 ± 1240 MFI, *p* = 0.042), but not of HLA-I (13,163 ± 1611 vs. 13,733 ± 1492 MFI, n.s.). The other HLA-C allotypes with the C2-epitope showed maintained or increased expression of HLA-C. Similar results were observed when patients with B-ALL were analyzed independently, with patients who carried the HLA-C*04 allotype showing a significant reduction in HLA-C expression (6903 ± 1412 vs. 12,026 ± 1581 MFI, *p* = 0.019) compared to patients with other C2-epitope allotypes (Figure 5).

Finally, since HLA C2-epitope is a ligand for KIR2DS1 as well, the role of this activating receptor in the outcome of acute leukemia pediatric patients was evaluated (Appendix A). The number of NK cells expressing KIR2DL1 (18.35% ± 1.48 vs. 18.34% ± 1.76) or KIR2DS1 (11.4% ± 1.07 vs. 11.4% ± 1.08) in the peripheral blood of patients with or without the KIR2DL1/HLA-C*04 interaction showed no differences (Appendix A). Furthermore, the presence of KIR2DS1 did not alter the differential EFS curves in patients with or without the KIR2DL1/HLA-C*04 interaction (Appendix A). KIR2DS1 did not add an independent prognostic value to the EFS of patients (HR = 1.373, *p* = 0.599).

### 2.4. KIR2DL1/C*04 Interaction Is an Independent Prognostic Biomarker That Complements Risk Status at Enrollment

Cox regression multivariate analysis of total acute leukemia patients showed that KIR2DL1/HLA-C*04 interaction was an independent prognostic factor for EFS (HR = 4.795, *p* = 0.005, Harrell C-statistic = 0.798 ± 0.051) and OS (HR = 6.731, *p* = 0.005, Harrell C-statistic = 0.837 ± 0.047), as was the risk status at enrollment (HR = 2.209, *p* = 0.013 for EFS and HR = 2.182, *p* = 0.033 for OS) when analyzed together with sex, age, the type of leukemia and the low/high expression of ULBP-1 on leukemic blast cells (Figure 6A).

Indeed, the presence of the KIR2DL1/C*04 interaction complemented the risk status at patient enrollment whether they were new diagnoses or relapses. Although similar suitable 5y-EFS rates were observed for standard-/intermediate-risk patients according to the absence/presence of the KIR2DL1/C*04 interaction (95.6% vs. 90.5%), a much lower 5y-EFS rate was observed in high-risk patients in the presence of the KIR2DL1/C*04 interaction (38.5% vs. 85.7%, *p* < 0.002) (Figure 6B). The negative prognostic value of the absence/presence of the KIR2DL1/C*04 interaction was maintained both in patients enrolled at diagnosis (5y-OS rate of 96.9% vs. 82.8%, *p* < 0.015) and in patients enrolled at relapse (5y-OS rate of 66.7% vs. 20.0%, n.s.), although in relapsed patients results were not statistically significant due to their reduced number (n = 8) (Figure 6C).

### 2.5. Models to Predict NK Cell Alloreactivity in Allogeneic HSCT in Our Series

Nineteen allogeneic HSCTs in 18 high-risk patients (7 AML, 8 B-ALL, and 3 T-ALL) were performed with 6 haploidentical, 5 related (RD), and 8 unrelated (UD) donors (Appendix A). Seven out of 19 HSCT patients relapsed (36.8%), and 6 died (31.5%). According to donor type, we found 66.7%, 40.0%, and 12.5% cumulative incidence of relapse and 50.0%, 20.0%, and 25% incidence of death for haploidentical, RD, and UD donors, respectively (Appendix A, up).

KIR2DL1/HLA-C*04 interaction was present in 4 out of 6 (66.7%) patients who relapsed and in 66.7% of their donors, as well as in 6 out of 6 (100%) patients who died and in 83.3% of their donors. It is important to bear in mind that the frequency of KIR2DL1/HLA-C*04 interaction in healthy controls from our series was 19.3%. Nonetheless, KIR2DL1/HLA-C*04 interaction was also present in 7 out of 12 (58.3%) patients without relapse and in 50.0% of their donors, as well as in 6 out of 13 (46.15%) patients who survived and in 38.4% of their donors.

According to models used to predict NK cell alloreactivity [53], the receptor-ligand model showed increasing cumulative incidence of relapse and death in donor-recipient pairs with missing ligands for KIR2DL1 (*n* = 1, 0% and 0%), KIR2DS1 (*n* = 4, 25% and 0%), KIR3DL2 (*n* = 13, 31% and 23%), KIR3DL1 (*n* = 13, 54% and 38%) and KIR2DL2/L3 (*n* = 4, 50% and 50%). For the ligand-incompatibility model increasing cumulative incidence of relapse and death were found in donor-recipient pairs with licensed KIR2DS1 (*n* = 4, 25% and 0%) and KIR3DL1 (*n* = 6, 50%, and 33%), with the worst results for those without licensed KIRs for missing ligands (*n* = 9, 44% and 44%). Although the small number of cases did not allow us to draw strong conclusions, donor-recipient pairs with functional KIR2DS1 (donor no C2-homozygous and recipient bearing its C2 ligands [29]) showed the best results (Appendix A, down).

## 3. Discussion

Maintenance therapy with 6-mercaptopurine and methotrexate is critical to ensure the cure of most cases of childhood acute lymphoblastic leukemia. The antileukemic effect of this therapy is primarily mediated by DNA incorporation of thioguanine nucleotides that favor nucleotide mismatching, which causes apoptosis after futile attempts of mismatch repair [54]. It is also suspected that immunological effectors may play a role in the eradication of tumor cells during this phase. Although these mechanisms are largely unknown, the high diversity of molecules guiding the interactions of NK cells with their tumor targets make them suitable candidates to play a role in tumor immune surveillance during this period, and it could help explain outcome variability among patients. In fact, NK cells are the first lymphoid cell to recover after therapy [55,56] and could be involved in the elimination of cancer cells. In support of this hypothesis, numerous studies have shown that HSCT with the appropriate haploidentical donor may improve clinical outcomes [8,9,57,58,59]. The main effectors in haploidentical HSCT are NK cells, which trigger their antitumor activity in case of missing ligand in the leukemic cells [8,9]. Our results show how the expression of ligands for the receptors that regulate NK cell function on leukemic cells is associated with the clinical outcome and survival of patients with ALL and AML. Interactions of these receptors, specifically KIR2DL1 with HLA-C*04 allotype (bearing the C2-epitope), have an independent predictive value and might complement the current risk stratification used in the clinical management of pediatric patients, particularly in those with high-risk features. This could contribute to personalizing not only the type of induction and consolidation therapy but also the intensity/length of maintenance therapy. Maintenance therapy, although less intensive, does have risks, as patients can develop life-threatening infections and late effects from prolonged cytotoxicity [60]. Therefore, children at lower immunological risk of events or death could benefit from reduced or, on the contrary, from increased maintenance treatment, whereas high immunological risk children could benefit from alternative therapies.

Recent evidence demonstrates that thiopurine-induced drug resistance mutations facilitate pediatric ALL relapses in a subset of patients, suggesting that decreasing exposure to these drugs by shortening the maintenance period might be beneficial because it reduces the chance of developing chemoresistant clones [61]. However, this reduction should be based on appropriate biomarkers because attempts to reduce the duration of maintenance therapy have generally led to an increase in relapses. Results from our series of pediatric acute leukemia show a clear association between relapse and the presence of the KIR2DL1/HLA-C*04 interaction, present in 17.6% of patients with relapse and only in 4.4% of patients without relapse, which highlights the possible participation of immune mechanisms in this process. Furthermore, our data reveal that leukemic cells of patients with the KIR2DL1/HLA-C*04 interaction have a selective loss in the expression of HLA-C but not of total HLA-I. Although the DT9 antibody used to assess the expression of HLA-C in our series may recognize both HLA-C and HLA-E; apparently, HLA-C is the antigen primarily detected by the antibody because HLA-E is poorly expressed in peripheral blood lymphocytes [62]. Unfortunately, our results do not allow us to determine whether the loss of HLA-C expression is causative of relapse and/or death of patients or if the surveillance pressure exerted by NK cells through the KIR2DL1/HLA-C*04 interaction could contribute to selecting more aggressive leukemic clones that specifically reduce the expression of HLA-C (and increase ligands for NK receptors such as ULBP-1) to escape the immune response. Our results do not allow us either to establish whether the effectors driving this phenomenon are NK cells or cytolytic T cells through the recognition of tumor peptides presented in HLA-C*04. This fact, however, seems less likely given that the negative effect of the KIR2DL1/HLA-C*04 interaction was observed in different lineages of acute leukemia, lymphoblastic (B and T), and myeloblastic. Yet, in turn, the protection that the KIR2DL1/C*05 interaction seems to exert against the detrimental effect of the KIR2DL1/HLA-C*04 interaction suggests the involvement of specific immunological mechanisms. In any case, these results are in line with those describing that peptide selectivity of KIR/Ligand interaction would allow NK cells to respond, not only to changes in the surface expression of HLA-C but also to the more subtle changes in the HLA-C peptidome [19].

Although the molecular mechanisms involved in the negative effect of the KIR2DL1/HLA-C*04 interaction need to be elucidated, there are important precedents showing that: (1) KIR2DL1/C2-epitope interaction is associated with increased susceptibility to childhood ALL and with an elevated risk of late relapse [63]; (2) frequency of HLA-Cw4 is elevated in patients with ALL (relative risk = 2.01, *p* < 0.0003), AML (relative risk = 2.06, *p* < 0.0002), and CML (relative risk = 2.14, *p* < 0.0008) [64]; (3) HLA-Cw4 is associated with ALL [65,66]; (4) C2-epitope is associated with reduced EFS in patients with B-CLL, particularly in KIR2DS1 negative patients, whose only receptor for the C2-epitope is KIR2DL1 [67]; and (5) presence of C2-epitope is associated with increased risk of relapse after HLA-matched transplantation in patients with acute and chronic myeloid leukemia, irrespective of donor relation [68]. Unfortunately, the aforementioned study did not evaluate the role of the C2-epitope at specific HLA-C allotype levels. Although the small number of patients who received HSCT in our series did not allow us to adequately explore the relationship of donor-derived KIR/HLA interactions with the patient’s bones, according to previous studies [68]. KIR2DL1/HLA-C*04 interaction was associated with poor outcomes, irrespective of the type of the allogenic HSCT (haploidentical, related, or unrelated) performed in our high-risk relapsed patients. Since the KIR2DL1/HLA-C*04 genotype seems to favor the escape of leukemic clones refractory to conventional and NK cell therapies (HSCT), these children should be candidates for alternative treatments such as BiTe, CART, or experimental treatment for AML. On the contrary, and although the associated mechanisms are unknown, children who carry the HLA-C*05 allotype seem to have more favorable outcomes and would be suitable candidates for NK cell therapy in the unlikely event of relapse, since relapse in children with this genotype is rare.

An independent study including most ALL pediatric patients from our series reported that persistent secondhand smoke (SHS) of smoking parents worsens OS and treatment-related mortality in these children [69]. Authors described that the main cause of death associated with SHS was an infection, which poses a new question for our study: could the KIR2DL1/HLA-C*04 interaction be related to defective protection against infections instead of to an altered antitumor response? However, since KIR2DL1/HLA-C*04 interaction was associated with a higher rate of relapse, immunosuppression caused by intensification treatments could be the main reason for infection and death. Nonetheless, to answer this question, future biological studies on a new and larger series of pediatric patients with acute leukemia will be necessary.

Regarding the expression of ligands for NK cell activating receptors, our results for ULBP-1, a ligand of NKG2D, are in disagreement with those reported for AML, in which a higher expression of ULBP-1 was associated with reduced relapse and improved 2-year overall survival [70]. This supports, in contrast, that sustained interaction of NKG2D receptor with its ligands expressed on tumor cells decreases NK cell-mediated cytolysis by uncoupling NKG2D receptor signaling from intracellular calcium mobilization [41]. However, in our series, these data had no independent prognostic value when compared with the risk at enrollment and the presence of the KIR2DL1/HLA-C*04 interaction. MICA/B expression shows marginal no significant changes between leukemic and normal cells. However, a higher expression of this NKG2D ligand was associated with reduced EFS, which is in agreement with the reduced NK cell function associated with reduced expression of activating NK receptors in high-risk myelodysplastic syndrome [71]. In contrast, the expression of DNAM-1 ligands (CD112 and CD155) showed the most significant increases in leukemic cells compared to normal cells, but this had no impact on the survival of patients. This suggests that, in pediatric acute leukemia, effective NK cell immune surveillance might depend on NKG2D more than on DNAM-1.

Collectively, our results show that the expression of ligands for the receptors that regulate the function of NK cells on leukemic cells is associated with the clinical outcome of patients with ALL and AML. Interactions of these receptors, specifically KIR2DL1 with HLA-C*04 allotype (bearing the C2-epitope), have an independent predictive value and might complement the current risk stratification used in the clinical management of pediatric patients. This could contribute to personalizing the type and intensity of therapy. Since the KIR2DL1/HLA-C*04 genotype seems to favor the escape of leukemic clones refractory to conventional and HSCT NK cell-based therapies, these children should be candidates for alternative treatments such as BiTe, CART, or experimental AML treatment.

## 4. Materials and Methods

### 4.1. Patients and Samples

This prospective observational study included 83 sex and aged-matched healthy controls and 102 consecutive (from 2012 to 2020) pediatric patients with B acute lymphoblastic leukemia (B-ALL, *n* = 70), T acute lymphoblastic leukemia (T-ALL, *n* = 16), and acute myeloid leukemia (AML, *n* = 16) from the Clinic University Hospital Virgen de la Arrixaca (Murcia, Spain) (Table 1). The institutional review board (IRB-00005712) approved the study. Written informed consent was obtained from all patients and controls (or their parents) in accordance with the Declaration of Helsinki.

EDTA anti-coagulated peripheral blood (PB) samples for HLA-I and KIR genotyping were obtained from healthy children randomly selected from unrelated volunteers, without chronic diseases, and with a negative family history of hereditary diseases, autoimmune diseases, or malignancies.

EDTA anti-coagulated BM samples were obtained from all patients, at diagnosis or at relapse, to perform cytomorphology, immunophenotype, cytogenetics (karyotype and fluorescence in situ hybridization, FISH), and molecular studies. KIR ligand expression was also evaluated on leukemic blast cells from these BM samples. Diagnostic criteria for the type and subtype of acute leukemia were based on the WHO classification of tumors of hematopoietic and lymphoid tissues [72]. Patient risk was estimated according to age, peripheral blood leukocyte count, immunophenotype, infiltration of central nervous system or testes, cytogenetic abnormalities, sensitivity to prednisone, and bone marrow response at day +15, +33, and +78. Following criteria of the SEHOP_PETHEMA_2013 protocol (https://www.sehh.es/images/stories/recursos/2014/documentos/guias/LAL_SEHOP_PETHEMA_2013.pdf (accessed on 15 January 2021)), ALL patients were classified into standard, intermediate or high risk. In AML patients, risk was classified as standard or high following the criteria of the NOPHO-DBH AML-2012 Protocol (https://www.skion.nl/workspace/uploads/NOPHO-DBH-AML-2012_Protocol-v2-1_17-01-2013.pdf (accessed on 15 January 2021)).

Depending on patient risk, ALL was treated according to the SEHOP-PETHEMA 2013 protocol or the previous PETHEMA 2005 protocol. In these protocols, all patients with T-ALL received L-asparagine depletion during maintenance and intensification/reinduction phases (https://www.fundacionpethema.es/2016/12/12/novedades-lal-sehop-pethema/ (accessed on 15 January 2021)). AML was treated according to the SHOP-LMA 2007 protocol or the most recent NOPHO-DBH AML 2012 protocol. For hematopoietic stem cell transplantation (HSCT), patients received a conditioning regimen followed by the infusion of donor cells. Patients remained hospitalized until hematopoietic and clinical recovery (details on Table 1).

### 4.2. KIR and HLA Genotyping

KIR and HLA-A, -B, and -C genotyping were performed in DNA samples extracted with QIAmpDNABlood Mini kit (QIAgen, GmbH, Hilden, Germany) using sequence-specific oligonucleotide PCR (PCR-SSO) and Luminex^®^ technology with Lifecodes KIR-SSO (cat. 545110R) and LifecodesHLA-SSO HLA-A (cat. 628913), HLA-B (cat. 628917) and HLA-C (cat. 628921) typing kits (Immucor Transplant Diagnostic, Inc. Stamford, CT, USA), as previously described [48,49]. HLA-A and HLA-B genotyping allowed us to identify alleles bearing the Bw4 motif according to the amino-acid sequences at positions 77–83 in the α1 domain of the HLA class I heavy chain. Bw4 alleles with threonine at amino acid 80 (80T) and higher affinity for KIR3DL1 (HLA-B*05, B*13, B*44) were distinguished from those with isoleucine 80 (I80) and lower affinity (HLA-A*23, A*24, A*25, A*32 and HLA-B*17, B*27, B*37, B*38, B*47, B*49, B*51, B*52, B*53, B*57, B*58, B*59, B*63, and B*77). HLA-C genotyping allowed distinction between HLA-C alleles with asparagine 80 (C1-epitope: HLA-C*01, 03, 07, 08, 12, 14, 16:01) and lysine 80 (C2-epitope: HLA-C*02, *04, *05, *06, *15, *16:02, *17, *18) [24]. Nonetheless, the KIR ligand calculator at https://www.ebi.ac.uk/ipd/kir/ligand.html (accessed on 15 January 2021) was used to ascertain Bw4, C1, and C2 epitopes.

KIR genotyping identified iKIRs (KIR2DL1, KIR2DL2, KIR2DL3, KRIR2DL5, KIR3DL1, KIR3DL2, and KIR3DL3) and aKIRs (KIR2DS1, KIR2DS2, KIR2DS3, KIR2DS4, KIR2DS5, and KIR3DS1), as well as KIR2DL4, which shows both inhibitory and activating potential. The method used could not distinguish between KIR2DL5A (telomeric) and KIR2DL5B (centromeric) forms. Different allotypes of KIR2DS4 were detected, including the expressed allotype KIR2DS4 full-length exon-5 (KIR2DS4f, KIR2DS4*001-002 alleles) and the non-expressed KIR2DS4 deleted exon-5 (KIR2DS4d, KIR2DS4*003, *004, *006, *007, and *008 alleles).

### 4.3. Immunophenotyping of Normal Cells and Leukemic Blasts

The expression of ligands for activating (MICA/B, ULBP-1, CD112, and CD155) and inhibitory (HLA-I and HLA-C) NK cell receptors were evaluated as mean fluorescence intensity (MFI) on BM normal residual cells and on leukemic blast cells using FACSCanto-II and BD FACSDiVa™ Software (Becton Dickinson, BD, San Diego, CA, USA). Photomultiplier (PMT) voltages were adjusted daily using CS&T beads (BD, Ref. 656047). Fluorescence compensations were adjusted every two months using blood samples stained for simple fluorochromes with conjugated anti-CD4 antibodies for each cytometer detector and then re-adjusted finely during the analysis using negative events as reference for each fluorochrome. Briefly, 100µL of BM samples diluted with PBS-1%BSA to contain 0.5 million total white cells were labeled in 5 tubes, all of them containing CD45-APCCy7 (BD, Ref. 348815) and, depending on the type of leukemia, CD7-PECy7 (BD, Ref. 564019) for T-ALL, CD19-PECy7 (BD, Ref. 341113) for B-ALL or CD34 PE-Cy7 (BD, Ref. 348811) or CD33-PECy7 (BD, Ref. 333946) for AML, so as to detect total lymphocytes, granulocytes and leukemic blast cells following the gating strategy described in Figure 7. Then, we added: (1) for tube-1, isotype-γ2a-FITC and isotype-γ1-PE (BD, Ref. 340394), isotype-γ2a-APC (Biolegend, Ref. 400219), and isotype-γ2b-BV421 (BD, Ref. 562748); (2) for tube-2, HLA-I-FITC (Bio-Rad, AbDSerotec, Kidlington, U.K., Ref. MCA81F); (3) for tube-3, purified anti-HLA-C (clone DT9, kindly provided by Dr. Simon Brackenridge from Oxford University, U.K.), which was revealed with a polyclonal anti-IgG (minimal x-reactivity)-BV421 (Biolegend, Ref. 405317) and then CD112 (Nectin-2)-PE (BD, Ref. 337410); (4) for tube-4, CD155 (PVR)-PE (BD, Ref. 337610); and (5) for tube-5, ULBP-1-PE (R&D Systems, Minneapolis, MN, USA, Ref. FAB1380P) and MICA/B-APC (BioLegend, San Diego, CA, USA, Ref. 320908).

MFI of lymphocytes, granulocytes, and leukemic blast cells in the FITC, PE, APC, and V450 (BV421) channels were estimated in the isotype tube, and these MFIs (background) were subtracted from the MFI values estimated for CD112, CD155, and ULBP-1 in the PE-channel, MICA/B in the APC-channel, HLA-I in the FITC-channel and HLA-C in the V450 channel for leukocyte subsets, respectively.

### 4.4. Statistical Analysis

All data were collected in a database (Excel2003; Microsoft Corporation, Seattle, WA, USA) and analyzed with IBM SPSS Statistics v15.0 (SPSS Inc., Chicago, IL, USA). Student t-test and ANOVA with post hoc tests were used to analyze continuous variables. Data were expressed as mean ± SEM. Kaplan–Meier and log-rank tests were used to analyze patient event-free survival (EFS) and overall survival (OS). Time to event or death was estimated in months from the date of enrollment. Multivariate analyses of prognostic factors for EFS and OS were performed using the Cox proportional hazards model (stepwise regression). Hazard ratio (HR) and 95% confidence interval were estimated. Harrell’s C-statistic was obtained using STATA-14 (Somersd package). *p* < 0.05 was considered statistically significant. The Bonferroni correction (Pc) was applied when needed.

## 5. Conclusions

Expression of ligands for activating and inhibitory receptors of NK cells on leukemic cells is associated with the clinical outcome of patients with ALL and AML. Specifically, reduced expression of HLA-C on leukemic cells of patients with the KIR2DL1/HLA-C*04 interaction was associated with a higher rate of relapse and lower patient survival. KIR2DL1/HLA-C*04 interaction was an independent prognostic factor that complemented the risk stratification used in the therapeutic management of pediatric patients.

## Figures and Tables

**Figure 1 cancers-13-02294-f001:**
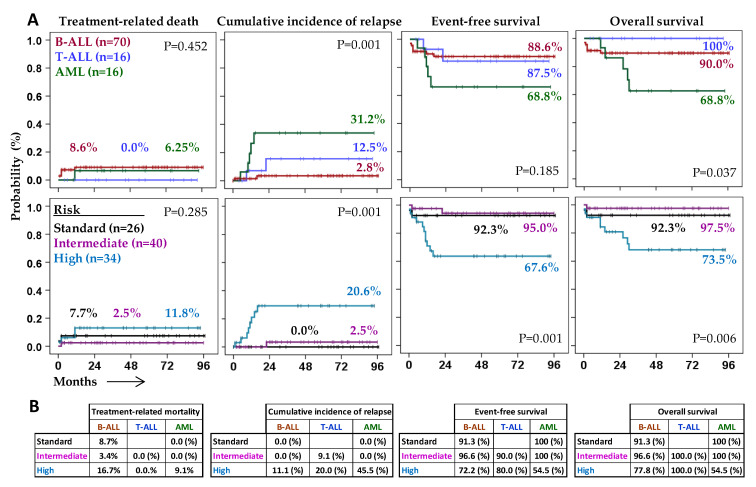
Patient outcome according to the type of acute leukemia and the risk status at enrollment. (**A**) Kaplan–Meier and log-rank tests for cumulative incidence of relapse, treatment-related mortality, event-free survival (EFS), and overall survival (OS) according to the type of acute leukemia (B-ALL, T-ALL, or AML) and the risk status at enrollment. (**B**) Rates (%) of cumulative incidence of relapse, treatment-related mortality, EFS, and OS according to the risk status of patients at diagnosis for each type of leukemia. Primary data can be consulted in Appendix A.

**Figure 2 cancers-13-02294-f002:**
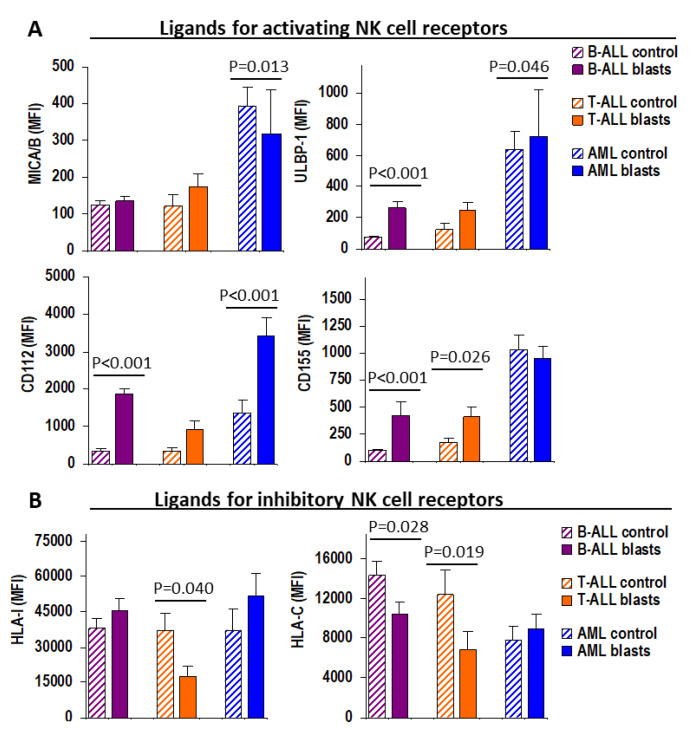
Expression of ligands for NK cell receptors on leukemic blast cells. (**A**) Mean fluorescence intensity (MFI) expression on normal counterpart cells (control, striped bars) and leukemic blast cells (solid bars) of ligands for activating NK cell receptors MICA/B and ULBP (ligands of NKG2D) and CD112 and CD155 (ligands of DNAM-1/CD226). (**B**) Expression of HLA-I and HLA-C ligands for inhibitory receptors (iKIRs and NKG2A). See Figure 7 for details of MFI estimation in leukocyte subset.

**Figure 3 cancers-13-02294-f003:**
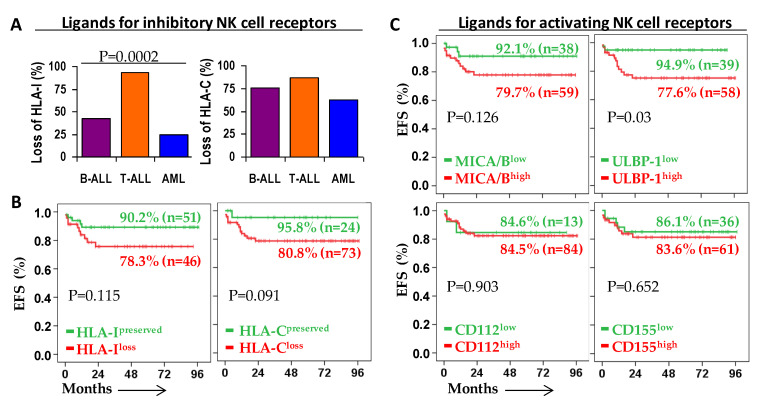
Event-free survival (EFS) of patients according to the expression of ligands for NK cell receptors. (**A**) Frequency of HLA-I and HLA-C loss in B-ALL, T-ALL and AML patients. HLA loss was computed when the mean fluorescence intensity (MFI) expression of HLA on leukemic blast cells was lower than that observed on the normal counterpart cells. P estimated with the Chi-squared test. (**B**) Kaplan–Meier and log-rank tests for EFS of acute leukemia patients according to the low or high expression of ligands for activating NK cell receptors MICA/B (ROC, AUC = 0.53, cutoff = 108.5, sensitivity = 80.0, and specificity = 51.0), ULBP-1 (ROC, AUC = 0.60, cutoff = 140.1, sensitivity = 86.6%, and specificity = 60.1%), CD112 (ROC, AUC = 0.55, cutoff = 1640, sensitivity = 66.0%, and specificity = 55.0%) and CD155 (ROC, AUC = 0.54, cutoff = 326.1, sensitivity = 56.0, and specificity = 62.2). (**C**) Kaplan–Meier and log-rank tests for EFS of acute leukemia patients according to the loss or preservation of HLA-I and HLA-C.

**Figure 4 cancers-13-02294-f004:**
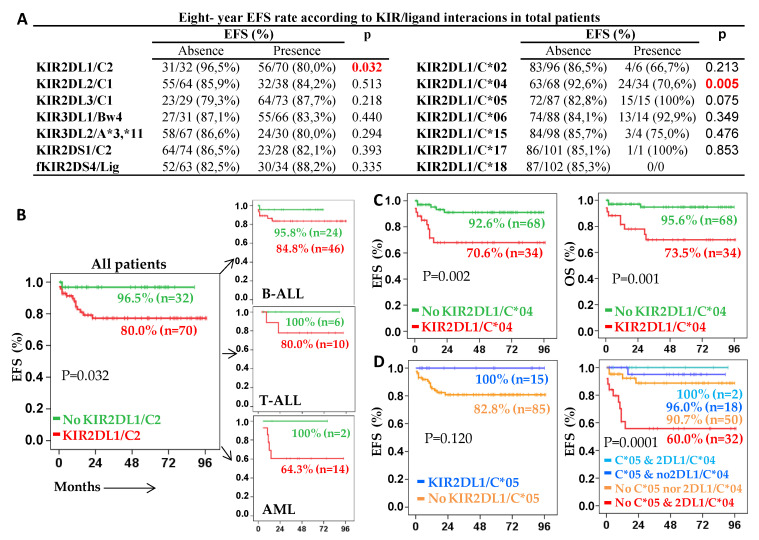
KIR2DL1/HLA-C2-epitope and specifically KIR2DL1/HLA-C*04 interactions are associated with reduced EFS and OS of acute leukemic pediatric patients. (**A**) Five-year event-free survival (5y-EFS) rates for KIR/HLA-ligand interaction (**left**) and for KIR/HLA-C allotypes with the C2-epitope (**right**). (**B**) Kaplan–Meier and log-rank tests for EFS of total, B-ALL, T-ALL and AML patients according to the presence of KIR2DL1/C2-epitope interaction. (**C**) Kaplan–Meier and log-rank tests for EFS and OS of acute leukemia patients according to the presence of KIR2DL1/HLA-C*04 interaction. (**D**) Kaplan–Meier and log-rank tests for EFS of acute leukemia patients according to the presence of KIR2DL1/HLA-C*05 interaction (**left**) or according to the combined presence of KIR2DL1/HLA-C*04 and KIR2DL1/HLA-C*05 interactions (**right**).

**Figure 5 cancers-13-02294-f005:**
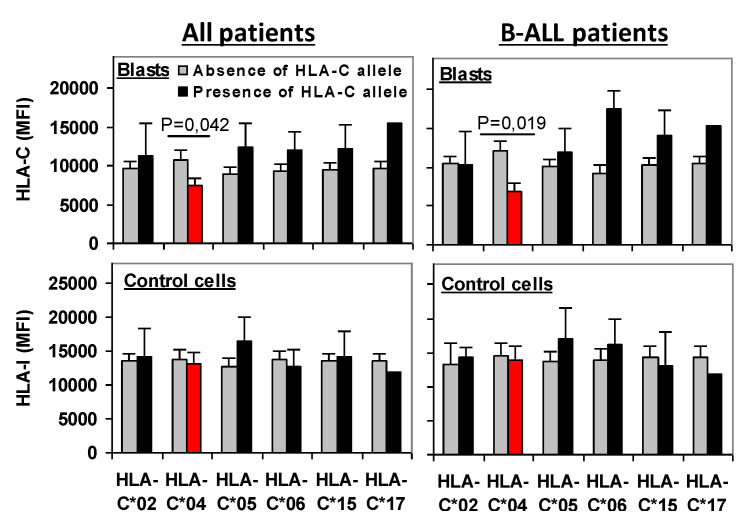
Loss of HLA-C expression was selectively observed in patients carrying the HLA-C*04 allotype. HLA-C and HLA-I expression on leukemic blast cells and on control cells in total acute leukemia patients and B-ALL patients carrying HLA-C allotypes with the C2-epitope. *p* estimated with Student’s *t*-test.

**Figure 6 cancers-13-02294-f006:**
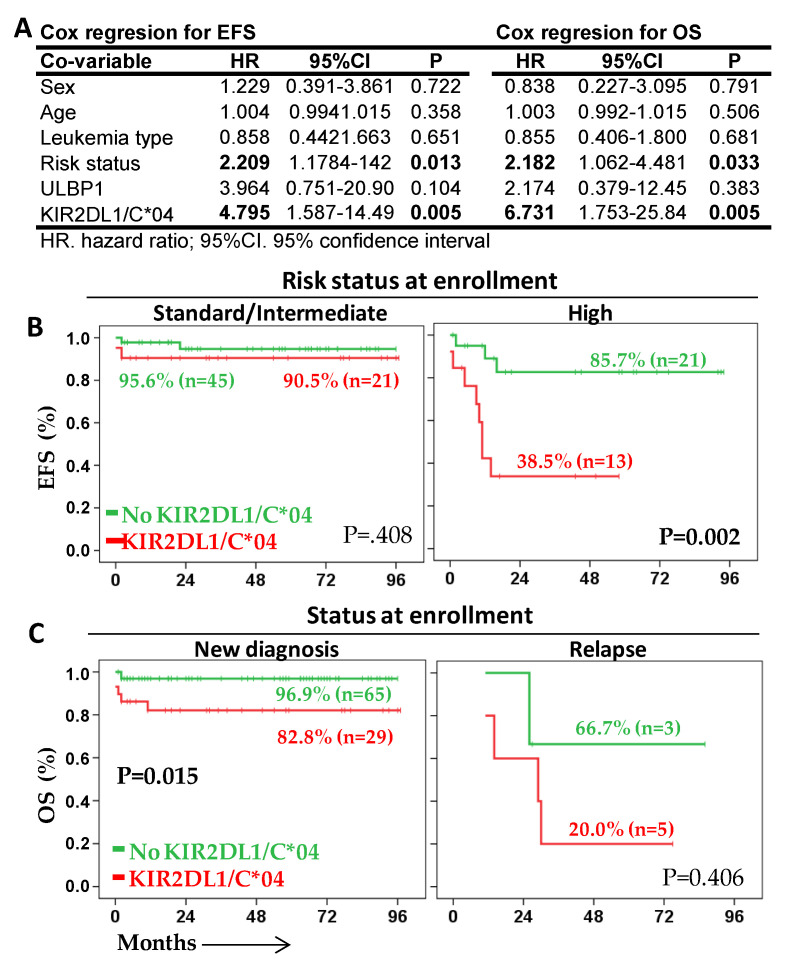
KIR2DL1/C*04 interaction is an independent prognostic biomarker. (**A**) Cox regression analysis for PFS and OS for sex, age, type of leukemia, risk status at enrollment, low/high expression of ULBP-1, and presence/absence of the KIR2DL1/C*04 interaction. (**B**) Kaplan–Meier and log-rank tests for EFS according to the presence/absence of KIR2DL1/HLA-C*04 interaction in standard/intermediate and high-risk patients. (**C**) Kaplan–Meier and log-rank tests for OS according to the presence/absence of KIR2DL1/HLA-C*04 interaction in patients enrolled because of new diagnosis or relapse.

**Figure 7 cancers-13-02294-f007:**
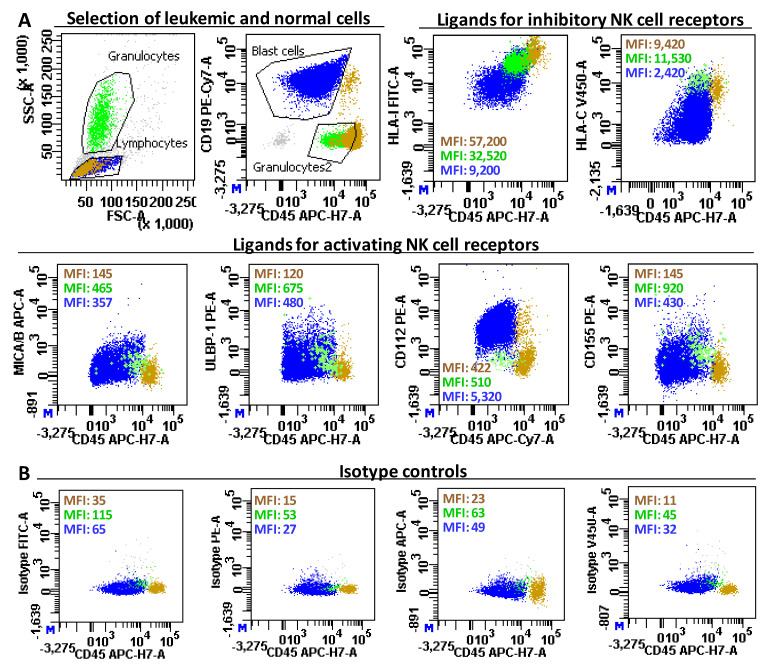
Immunophenotype analysis of NK cell receptor ligands in normal and leukemic bone marrow cells. (**A**) Events with low/intermediate dispersion in FSC/SSC, including residual normal lymphocytes (brown dots) and leukemic blast cells (blue dots), were gated in an FSC/SSC dot plot. Leukemic cells were gated in a plot with CD45-APCCy7 combined with CD19-PECy7 for B-ALL, CD7-PECy7 for T-ALL or CD34-PECy7 or CD33-PECy7 for AML. Granulocytes (green dots) were identified with gates on high SCC in the FSC/SSC dot plot and low CD45 in the CD45-APCCy7/PECy7 dot plot. Mean fluorescence intensity (MFI) was evaluated for CD112, CD155, and ULBP-1 in the PE-channel, for MICA/B in the APC-channel, for HLA-I in the FITC-channel, and for HLA-C in the V450 (VB421) channel on lymphocytes, granulocytes, and blast cells. Dot-plots correspond to a representative case with a clear expression loss of HLA-I and HLA-C and strong gain of CD112, with a moderate gain of MICA/B, ULBP-1, and CD155. (**B**) MFI in the FITC, PE, APC, and V450 (BV421) channels were also estimated for leukocyte subsets in the isotype tube to determine the background of each fluorescence.

**Table 1 cancers-13-02294-t001:** Biological, clinical, therapeutic, and evolutionary features of patients.

	Total (*n* = 102)	B-ALL (*n* = 70)	T-ALL (*n* = 16)	AML (*n* = 16)
Age (month, mean ± SD)	83.0 ± 49.3	82.5 ± 49.1	91.7 ± 44.9	76.6 ± 55.6
Sex (male, *n* (%))	63 (61.8%)	42 (60.0%)	12 (75.0%)	9 (56.3%)
Inclusion in the study				
At diagnosis, *n* (%)	96 (94.1%)	66 (94.3%)	16 (100%)	14 (87.5%)
At relapse, *n* (%)	6 (5.9%)	4 (5.7%)		2 (12.5%)
PB-WBC count at diagnosis				
<10 × 10^3^/µL	89 (87.3%)	65 (92.9%)	10 (62.5%)	14 (87.5%)
10–50 × 10^3^/µL	5 (4.9%)	4 (5.7%)		1 (6.3%)
>50 × 10^3^/µL	8 (7.8%)	1 (1.4%)	6 (37.5%)	1 (6.3%)
Risk stratification ^1^				
Standard, *n* (%)	27 (26.4%)	23 (32.9%)	0 (0.0%)	3 (18.7%)
Intermediate, *n* (%)	41 (40.2%)	29 (41.1%)	11 (68.7%)	2 (12.5%)
High, *n* (%)	34 (34.0%)	18 (25.7%)	5 (31.3%)	11 (68.8%)
Treatments				
PETHEMA-96/2001/2005	13 (12.6%)	12 (17.0%)	1 (6.2%)	
LAL-SEHOP-PETHEMA-2013	73 (71.4%)	58 (82.7%)	15 (93.7%)	
LMA-SHOP-2007	10 (9.8%)			10 (62.5%)
NOPHO-DBH-AML-2012	6 (5.8%)			6 (37.5%)
HSCT, *n* (%)	18 (17.6%)	8 (11.4%)	3 (18.8%)	7 (43.8%)
Allo-HSCT, *n* (%)	13 (12.7%)	5 (7.1%)	3 (18.8%)	5 (31.3%)
Related, *n* (%)	5 (38.5%)	2 (40.0%)	2 (66.7%)	3 (42.9%)
Unrelated, *n* (%)	8 (61.5%)	3 (60.0%)	1 (33.3%)	4 (57.1%)
Haplo-HSCT, *n* (%)	5 (4.9%)	3 (4.3%)		2 (12.5%)
Clinical outcome				
Mean follow-up (months)	41.98 ± 31.5	41.9 ± 32.9	46.6 ± 30.6	37.7 ± 25.9
Relapse, *n* (%)	9 (8.8%)	2 (2.8%)	2 (12.5%)	5 (31.3%)
Death, *n* (%)	12 (11.8%)	7 (10.0%)		5 (31.3%)

^1^ Risk stratification: according to specific criteria (described in method section); B-ALL: B-cell acute lymphoblastic leukemia; T-ALL: T-cell acute lymphoblastic leukemia; AML: acute myeloid leukemia; SD: standard deviation; BM: bone marrow; PB: peripheral blood; HSCT: hematopoietic stem cell transplantation; EFS: event-free survival; OS: overall survival; control group (*n* = 83): age 74.4 ± 35.7 months. Male *n* = 47 (56.6%); female *n* = 36 (43.4%).

**Table 2 cancers-13-02294-t002:** KIR gene, HLA-I ligand, and KIR/HLA-I ligand interaction frequencies in controls and patients.

KIR Genes, *n* (%)	Controls (*n* = 83)	Patients (*n* = 102)	B-ALL (*n* = 70)	T-ALL (*n* = 16)	AML (*n* = 16)
KIR2DL1	80 (96.4%)	99 (97.1%)	68 (97.1%)	16 (100%)	15 (93.8%)
KIR2DL2/2DS2 ^1^	48 (57.8%)	49 (48.0%)	32 (45.7%)	7 (43.8%)	10 (62.5%)
KIR2DL3	74 (89.2%)	93 (91.2%)	64 (91.4%)	15 (93.8%)	14 (87.5%)
KIR2DL5	44 (53.0%)	56 (54.9%)	35 (50.0%)	8 (50.0%)	13 (81.3%)
KIR3DL1	81 (97.6%)	100 (98.0%)	68 (97.1%)	16 (100%)	16 (100%)
KIR2DS1	30 (36.1%)	38 (37.3%)	23 (32.9%)	7 (43.8%)	8 (50.0%)
KIR2DS3	26 (31.3%)	35 (34.3%)	21 (30.0%)	5 (31.3%)	9 (56.3%)
KIR2DS4full	35 (42.2%)	54 (52.9%)	34 (48.6%)	11 (68.8%)	9 (56.3%)
KIR2DS5	26 (31.3%)	30 (29.4%)	19 (27.1%)	6 (37.5%)	5 (31.3%)
KIR3DS1	29 (34.9%)	38 (37.3%)	24 (34.3%)	7 (43.8%)	7 (43.8%)
**HLA-I ligands, *n* (%)**					
HLA-A*03/A*11	23 (27.7%)	30 (30.6%)	20 (30.8%)	4 (25.0%)	6 (37.5%)
Bw4 epitope	65 (78.3%)	70 (68.6%)	49 (70.0%)	10 (62.5%)	11 (68.8%)
Bw4-80T	20 (24.1%)	22 (21.6%)	16 (22.9%)	4 (25.0%)	2 (12.5%)
Bw4-80I	53 (63.9%)	61 (59.8%)	44 (62.9%)	8 (50.0%)	9 (56.3.3%)
HLA-C1 epitope	61 (73.5%)	80 (78.4%)	56 (80.0%)	13 (81.3%)	11 (68.8%)
HLA-C2 epitope	56 (67.5%)	73 (71.6%)	48 (68.6%)	10 (62.5%)	15 (93.8%)
KIR2DS4-ligands ^2^	44 (53.0%)	71 (71.0%) ^3^	49 (72.1%)	10 (62.5%)	12 (75.0%)
**KIR/HLA-ligand interactions, *n* (%)**					
KIR2DL1/C2	54 (65.1%)	70 (68.6%)	46 (65.7%)	10 (62.5%)	14 (87.5%)
KIR2DL2/C1	34 (41.0%)	38 (37.3%)	27 (38.6%)	5 (31.3%)	6 (37.5%)
KIR2DL3/C1	54 (65.1%)	73 (71.6%)	52 (74.3%)	12 (75.0%)	9 (56.3%)
KIR3DL1/Bw4	63 (75.9%)	66 (68.0%)	45 (69.2%)	10 (62.5%)	11 (68.8%)
KIR2DS1/C2	22 (26.5%)	28 (27.5%)	16 (22.9%)	5 (31.3%)	7 (43.8%)
KIR2DS4/Ligands ^1^	18 (21.7%)	33 (35.1%)	23 (35.4%)	5 (31.1%)	6 (37.5%)
KIR2DS4/C*04:01	7 (8.4%)	15 (15.5%)	10 (15.4%)	2 (15.5%)	3 (18.8%)
KIR3DS1/B*51	3 (3.6%)	4 (3.92%)	2 (2.85%)	2 (12.5%)	0 (0.0%)

^1^ KIR2DL2 and KIR2DS2 had a high linkage disequilibrium and showed the same frequency; ^2^ ligands for KIR2DS4: HLA-A*11, C*01:02, C*02:02, C*04:01, C*05:01, C*14:02, and C*16:01 [52]. ^3^ Controls vs. patients *p* = 0.009, Pc = 0.045.

## Data Availability

The data presented in this study are openly available in Appendix A.

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
