# Peer review of "Expression of NK Cell Receptor Ligands on Leukemic Cells Is Associated with the Outcome of Childhood Acute Leukemia"

_cancers, 2021, doi:10.3390/cancers13102294_

Round 1

Reviewer 1 Report

Systematic analyses of the expression of ligands for activating receptors in conjunction with detailed genetic analyses of the KIR/HLA backgrounds in the context of ALL and AML are largely lacking and crucial to understand the contribution of NK cells to the control of such malignancies and to use them therapeutically. As such the study presented here addresses a clear need in the field.

There are however concerns about the manuscript as presented.  Firstly, the number of patients in the AML and ALL- T groups while potentially quite informative for assessment of NK receptor ligand expression, seems small for any impactful data on survival/outcomes.  The ALL-B cohort is probably at about the limit if one is to explore KIR/HLA genetics with any degree of confidence.

The presentation of data around the expression of NK cell receptor ligands is not that robust.  While there is primary data showing the identification of healthy cells and leukemic blasts,  the representative data of the actual expression of these ligands on their cell surface needs improvement to make clear how the MFI data was generated for Fig 2

Exploring the 8y-EFS is logical if the cohort size is reasonable. However, it is unclear how the cutoffs for the stratification were determined…again primary data here would be helpful.  The extent to which data from all leukemias should be pooled is unclear. 

Similar to the activating ligand receptor expression, representative patterns of HLA loss should be shown.

Definition of KIR ligands is not straightforward and some further detail around how ligands have been determined/assigned is required.  For example it is clear that not all Bw4 alleles are ligands for KIR3DL1 and there is significant variation in the quality of different C1/C2 -KIR interactions that may well be clinically relevant.  The extent to which KIR3DS1 interacts with bw4 alleles is unclear/doubtful.  Nevertheless, the extent to which there is a clinical signal associated with C*04 is interesting.  It would be reassuring to see data on KIR2DS1-ve individuals who possess C*04 however the extent to which the study is powered for this analysis is unclear.  Similarly, data showing the frequency of KIR2DL1 expression and or the functional potential of cells expressing this receptor in C*04 vs non C*04 patients would be informative.

It is also worth pointing out that the DT9 mAb used to recognise HLA-C expression in this study also recognises HLA-E. Do the authors have measures of HLA-E expression independently that would verify that changes seen with DT9 do in fact reflect alterations to HLA-C expression?

Overall, I think the data set is interesting and worthy of publication. There are however many threads in this story and both the presentation of the data and the clarity of the narrative could be improved.

Author Response

To all reviewer: the manuscript has undergone a new professional language editing. Besides, several mistakes in the titles of figures and tables have been detected and corrected.

 Reviewer-1

Systematic analyses of the expression of ligands for activating receptors in conjunction with detailed genetic analyses of the KIR/HLA backgrounds in the context of ALL and AML are largely lacking and crucial to understand the contribution of NK cells to the control of such malignancies and to use them therapeutically. As such the study presented here addresses a clear need in the field.

There are however concerns about the manuscript as presented.  Firstly, the number of patients in the AML and ALL- T groups while potentially quite informative for assessment of NK receptor ligand expression, seems small for any impactful data on survival/outcomes.  The ALL-B cohort is probably at about the limit if one is to explore KIR/HLA genetics with any degree of confidence.

The presentation of data around the expression of NK cell receptor ligands is not that robust.  While there is primary data showing the identification of healthy cells and leukemic blasts, the representative data of the actual expression of these ligands on their cell surface needs improvement to make clear how the MFI data was generated for Fig 2

We are really sorry that the explanation of the immunophenotype analysis was not sufficient. From your comments we appreciate that it is convenient that we give a clearer explanation about how we conducted this analysis. You can see that we have improved the description of the methods in section 4.3 and in figure-7.

We have described more clearly how MFI was estimated for each ligand in each leukocyte subset, and how the background fluorescence was subtracted in each leukocyte subset using the MFI of each fluorochrome determined in the isotype tube. We have also better described not only how we adjusted compensations on a daily basis (fine tuning during the analysis) but also how we adjusted the instrument in the long term (bimonthly).

Maybe in the figure you are missing the gates to calculate %s of ligands in each leukocyte subset, but since HLA molecules were normally expressed in 100% cells, %s were not informative in all cases to show how the expression of these molecules was modulated. So we decided to work with the MFI of ligands, not with %s. Therefore, we did the analysis without gates on each ligand, so they are not shown in the figure. Once we selected the leukocyte subset as shown in figure-7 we recorded the MFI for each fluorescence channel for the total cells of each leukocyte subset. We have up-graded figure-7 to include plots showing the staining with the Isotype controls.

Exploring the 8y-EFS is logical if the cohort size is reasonable. However, it is unclear how the cutoffs for the stratification were determined…again primary data here would be helpful.  The extent to which data from all leukemias should be pooled is unclear.

We decided to describe 8-year survivals because lines of all analyzed groups reached 8 year in the graphs, but it is true that if we take into account the mean follow-up of our series (40 months) and the reduced number of patients in some groups, it would be more convenient and standardized to show 5-year survivals. Anyway, you would see that numbers did not change because all events and deaths took place in the first four/five years in our series.

We will be adding a supplementary Excel file with the primary actuarial and experimental data of patients included in this study.

Similar to the activating ligand receptor expression, representative patterns of HLA loss should be shown.

We are very sorry, but we are not sure what your suggestion in this comment is. Is it duplicating the legend of figure-2A in figure-2B, as shown below?

I cannot paste the figure here. Could you please have a look to the new-figure-2?

Definition of KIR ligands is not straightforward and some further detail around how ligands have been determined/assigned is required.  For example it is clear that not all Bw4 alleles are ligands for KIR3DL1 and there is significant variation in the quality of different C1/C2 -KIR interactions that may well be clinically relevant.  The extent to which KIR3DS1 interacts with bw4 alleles is unclear/doubtful.  Nevertheless, the extent to which there is a clinical signal associated with C*04 is interesting.

First, thank you very much for these suggestions that will clearly improve the manuscript. We have reviewed recent references to re-write the introduction and method sections.  In the introduction we have up-dated the KIR/Ligand interaction based on recent review. In

Methods (4.2 section) we have described which alleles were considered C1, C2, Bw4, Bw-80T and BW4-80I and indicated that the KIR ligand calculator was used to ascertain Bw4, C1 and C2 epitopes (https://www.ebi.ac.uk/ipd/kir/ligand.html)”

Second, it is true that two rare HLA-B alleles (B*46:01 and B*73:01) can be C1-ligands for KIR2DL2/L3. However, as you can see (http://www.allelefrequencies.net) the frequency of these alleles is very low (<0.0005%) in Spain or Europe. In fact we did not have any patient with these alleles as you can see in table-S1. So this fact does not change the results or conclusions of the manuscript.

Third, as described by Slawomir Gwozdowicz in 2019, we agree that affinity of each HLA allotype for their correspondent KIR can be very different, which could have clinical implications. Unfortunately, affinity information is only available for KIR:HLA-ligand pairs with available crystallographic data. Nonetheless, in our series only HLA-C*04 but no other HLA alleles with the C2-epitope had a clear impact on patient outcome. Although we believe that this could also be related to tumor peptidome presented on HLA (these facts have been described in the introduction, and discussed accordingly).

Fourth, we have reviewed our Bw4 data. We have classified Bw4 alleles in BW4-80T (with lower affinity for 3DL1) and BW4-80I (with higher affinity for 3DL1) and no impact on the outcome of acute leukemia patients was observed.

Please take a look to the figure I enclosed in the point-by-point word doc. I can not past the graph here.

Here, it should be a Kaplan-Meier analysis of EFS according to the presence or Bw4-80T. No differences.

Here, it should be a Kaplan-Meier analysis of EFS according to the presence or Bw4-80I. No differences.

No differences in the frequency of these 2 BW4-epitope isoforms were observed among study groups, either. Have a look to the new Table-2 of the manuscript.

We have included data for Bw4 80T and 80I isoforms in table-2.

We agree that the interaction KIR3DS1/Bw4 is controversial nowadays. KIR3DS1 ligands have remained indefinite until recently, and they are represented by the open conformation of the non-classical HLA-F molecule and by the HLA-B*51, an HLA-B Bw4 I80 allotype (revised in the reference 18 of the manuscript).  We have clearly stated this in the introduction and corrected Table-1 and results description, accordingly. We have also deleted KIR3DS1/Bw4 interaction from figure-4A.

It would be reassuring to see data on KIR2DS1-ve individuals who possess C*04 however the extent to which the study is powered for this analysis is unclear.  Similarly, data showing the frequency of KIR2DL1 expression and or the functional potential of cells expressing this receptor in C*04 vs non C*04 patients would be informative.

We have included a Supplementary Figure S2 with data of NK cells expressing either KIR2DL1 or KIR2DS1 in the peripheral blood of these patients. As you can see, no differences in the frequency of NK cells expressing the activating or the inhibitory receptor for the C2-epitope were observed. Besides, in the figure you can also see that the presence of the gene KIR2DS1 did not alter the EFS curves of patients according to the presence of the KIR2DL1/C*04 interaction. KIR2DS1 did not add any independent prognostic value to the EFS of patients (HR=1.373, P=0.599).

Since this is supplementary information, we have not added the reagents and the gaiting used for this flow cytometry analysis in the method section. We are now working on a new manuscript to show the expression of KIR receptors (%s and MFI) in this series of patients, comparing expression at diagnosis and at different times during the therapy. We will try to correlate the variations in the expression of KIR receptors in NK cells with that of their ligands in the blasts (in presence or absence of the leukemic cells). We are still analyzing data from the follow-up. We hope these data will be available for publication within the following months.

It is also worth pointing out that the DT9 mAb used to recognise HLA-C expression in this study also recognises HLA-E. Do the authors have measures of HLA-E expression independently that would verify that changes seen with DT9 do in fact reflect alterations to HLA-C expression?

We did not use any further antibody to ascertain HLA-C vs. HLA-E staining.

We know that a controversy exists about what DT9 antibody is detecteding in peripheral blood cells. But apparently there is a strong claim that “HLA-C is the antigen primarily detected by flow cytometry of peripheral blood lymphocytes with DT9”, since HLA-E expression is poorly detected (Science. 2013 Sep 13; 341(6151): 1175.;  doi: 10.1126/science.1241854).” We have discussed this fact accordingly and included the reference.

Overall, I think the data set is interesting and worthy of publication. There are however many threads in this story and both the presentation of the data and the clarity of the narrative could be improved.

Thanks.

Reviewer 2 Report

The authors have performed studies on the expression of NK cell receptors and ligands in relation to clinical parameters in a patient material from a cohort of 102 children diagnosed with acute leukemia. They find that KIR2DL1/HLA-C*04 expression present in a subgroup of cases is associated with a higher rate of relapse and lower patient survival. The manuscript is clearly written but would benefit from additional spell-checking of the text and figures.

Specific comments

NK cells are responsive to a multitude of ligands with corresponding receptors. It is not clear why the authors chose the particular ligands they have studied in the manuscript. This should be explained in more detail in the introduction.

The overall survival of the T-ALL group seems to be better than in other studies. Is this due to the small sample size or what is the authors explanation for this?

Figure 1 lacks p-values for some of the items and it is not clear if the p-values were corrected for multiple testing or not. Please add/indicate this.

Figure 1: “Event-free survival” is misspelled.

The coloring in Figure 1 is confusing. The High risk group should be indicated more clearly.

Figure 5: The p-values shown should be corrected for multiple testing.

Author Response

To all reviewer: the manuscript has undergone a new professional language editing. Besides, several mistakes in the titles of figures and tables have been detected and corrected.

Reviewer-2

The authors have performed studies on the expression of NK cell receptors and ligands in relation to clinical parameters in a patient material from a cohort of 102 children diagnosed with acute leukemia. They find that KIR2DL1/HLA-C*04 expression present in a subgroup of cases is associated with a higher rate of relapse and lower patient survival. The manuscript is clearly written but would benefit from additional spell-checking of the text and figures.

 Specific comments

NK cells are responsive to a multitude of ligands with corresponding receptors. It is not clear why the authors chose the particular ligands they have studied in the manuscript. This should be explained in more detail in the introduction.

We have extended the last paragraph of the introduction to add a brief explanation about why we aimed our studies at these particular ligands and not others. As you can see, we have referenced several of our previous publications where we have investigated the role of potential KIR/ligand or NKG2D interactions in the education of NK cells and in the immune surveillance of different types of tumors. Although in those works we monitored the expression of NK cell receptors (KIRs, NKG2A and DNAM-I) in a large series of patients, the expression of their ligands was monitored only in a reduced number of myeloma patients. Nonetheless, in this particular work the high expression of HLA-I and HLA-C in the myelomatous cells gave us the clue to understand the role of the genetic polymorphism of KIR/ligand interactions in this cancer. Besides, we described the important role of CD226 in the immune surveillance of solid tumors (melanoma, bladder and ovarian cancers) and the role of NKG2D polymorphism melanoma, so we were interested in the role of ligands for KIR receptors (HLA), DNAM-I (CD112 and CD155) and NKG2D (MICA/B and ULBP-1). Anyway, you have to understand that this prospective study performed with fresh leukemic cells started in 2012, so we decided at that time to analyze these specific ligands since they were the best defined NK cell ligands at that very moment.

The overall survival of the T-ALL group seems to be better than in other studies. Is this due to the small sample size or what is the authors explanation for this?

Although there is not a single clear and definite explanation, the small number of patients might partially explain it. However, the SEHOP/PETHEMA-2013 guidelines (https://www.fundacionpethema.es/2016/12/12/novedades-lal-sehop-pethema/) scheduled an intensification with L-asparragine depletion for intermediate and high risk group, where all T-ALL cases are allocated, that could have contributed to such a good results. To this regard, an additional comment is added within section 4.1 (patients and samples). In this series, two T-ALL patients relapsed, both were isolated extramedullary relapses, rescued with second-line treatment including allogeneic stem cell transplantation.

Figure 1 lacks p-values for some of the items and it is not clear if the p-values were corrected for multiple testing or not. Please add/indicate this.

Sorry, P-values were hidden behind the graphs when I pasted a new version of them.

Corrected; thank you.

However, we have consulted with our statistician and they say that in this survival graph we have used the log-rank test for one factor, without two-to-two comparisons, therefore no adjustment is necessary.

Figure 1: “Event-free survival” is misspelled.

Corrected; thank you.

 The coloring in Figure 1 is confusing. The High risk group should be indicated more clearly.

Corrected; thank you.

Figure 5: The p-values shown should be corrected for multiple testing.

Again, we have consulted with our statistician and they say that in the bar graph we are comparing presence-absence of one factor, it is a dichotomous variable and therefore it is not necessary to adjust.

Reviewer 3 Report

The authors showed the data concerning "Expression of ligands for NK cell receptors is associated with the outcome of childhood acute leukemia.". The hypothesis that  L/R profile in tumor  and NK cells  is deregulated in pediatric or adult cancers in general is already well described, and in particular for AML and ALL, and was clearly revised by the authors in introduction and discussion paragraph. Nevertheless, even if the cohort of patients is important in this study presented by the authors, there is so difficult to follow the results and clinical correlation are not clearly described.

Technical points: the methodology of all of flowcytometry data analysis in Ligand and Receptor expression is not optimal for different point of view:

  • firstly, the measurement of MFI and not RatioMFI based on negative control reflected a lot of nonspecific and auto fluorescent background, and misinterpreted data; the most appropriate control in most of case is FMO or Isotype Ctr in a separated tube, or Normal bone Marrow 
  • secondly, using the normal counterpart as neutrophil and lymphocytes to interpreted level of positivity in blasts of AML and ALL is not appropriated.

Clinical points: for AML and ALL in pediatric patients the OS and EFF are quite different, using entirely cohort of all this patients in survival curves is not a good reflect of ULBP1 or KIR2DL1 results, even if a significance in univariate analysis could be observed but not sufficient to translate as survival correlation parameter.

Minor Points: some English failed in translation on table nr 1 B-ALL , T-ALL and AML should be corrected

Author Response

To all reviewer: the manuscript has undergone a new professional language editing. Besides, several mistakes in the titles of figures and tables have been detected and corrected.

Reviewer-3

The authors showed the data concerning "Expression of ligands for NK cell receptors is associated with the outcome of childhood acute leukemia.". The hypothesis that  L/R profile in tumor  and NK cells  is deregulated in pediatric or adult cancers in general is already well described, and in particular for AML and ALL, and was clearly revised by the authors in introduction and discussion paragraph. Nevertheless, even if the cohort of patients is important in this study presented by the authors, there is so difficult to follow the results and clinical correlation are not clearly described.

Technical points: the methodology of all of flow cytometry data analysis in Ligand and Receptor expression is not optimal for different point of view:

firstly, the measurement of MFI and not RatioMFI based on negative control reflected a lot of non-specific and auto fluorescent background, and misinterpreted data; the most appropriate control in most of case is FMO or Isotype Ctr in a separated tube, or Normal bone Marrow

These suggestions are similar to the first suggestions of Reviewer #1(See the attachment). Could you please have a look at the response offered to that reviewer?

secondly, using the normal counterpart as neutrophil and lymphocytes to interpreted level of positivity in blasts of AML and ALL is not appropriated.

Also answered in the first suggestion of Reviewer #1 (See the attachment). And corrections done in the manuscript, accordingly.

Clinical points: for AML and ALL in pediatric patients the OS and EFF are quite different, using entirely cohort of all this patients in survival curves is not a good reflect of ULBP1 or KIR2DL1 results, even if a significance in univariate analysis could be observed but not sufficient to translate as survival correlation parameter.

We totally agree that actuarial data for ALL and AML are quite different in our series (Fig 1) and other series. That is why we did try to give information for each type of leukemia whenever we believed it was relevant (Table-1, Fig-1, Fig-2, Table-S1, Table-2, and Fig-4). Besides, we have included the type of leukemia in all Multivariate Cox-regression analysis performed all over the manuscript to correct possible bias.

In the particular case of ULBP1, I am pleased to offer you this information separately for ALL (B and T) and AML. As you can see results are quite similar in both cases but the results were not significant when studied separately. If you still believe that the message of the manuscript will improve by adding this information, please let us know and we will be pleased to prepare a supplementary figure with these data.

Please take a look at the point-by-point word document enclosed to this re-submission. 

Here, should be a Kaplan-Meier plot showing EFS curves for patients with low and high ULBP for Lymphoblastic leukemia. Curves are similar to the curves shown in the manuscript.

Here, should be a Kaplan-Meier plot showing EFS curves for patients with low and high ULBP for myeloblastic leukemia. Curves are similar to the curves shown in the manuscript, and similar to those of lymphoid leukemia.

In the case of KIR2DL1, data for each type of leukemia are shown in figure-4, or described in the manuscript “Patients with the KIR2DL1/HLA-C*04 interaction showed the most reduced 5y-EFS rates also in B-ALL (77.3% vs. 93.8%, p=0.036), T-ALL (75.0% vs. 91.7%, n.s.) and AML (50.0% vs. 87.5%, p=0.05) patients”.

Minor Points: some English failed in translation on table nr 1 B-ALL , T-ALL and AML should be corrected

Corrected; thank you.

Round 2

Reviewer 2 Report

I am satisfied with the revisions made by the authors.